# SuperCLIP: CLIP with Simple Classification Supervision

**Weiheng Zhao**[1]    **Zilong Huang**[2]*    **Jiashi Feng**[2]    **Xinggang Wang**[1]

[1]*School of EIC, Huazhong University of Science and Technology*
[2]*ByteDance*

Code & Models: hustvl/SuperCLIP

## Abstract

Contrastive Language-Image Pretraining (CLIP) achieves strong generalization in vision-language tasks by aligning images and texts in a shared embedding space. However, recent findings show that CLIP-like models still underutilize fine-grained semantic signals in text, and this issue becomes even more pronounced when dealing with long and detailed captions. This stems from CLIP's training objective, which optimizes only global image-text similarity and overlooks token-level supervision—limiting its ability to achieve fine-grained visual-text alignment. To address this, we propose SuperCLIP, a simple yet effective framework that augments contrastive learning with classification-based supervision. By adding only a lightweight linear layer to the vision encoder, SuperCLIP leverages token-level cues to enhance visual-textual alignment — with just a 0.077% increase in total FLOPs, and no need for additional annotated data. Experiments show that SuperCLIP consistently improves zero-shot classification, image-text retrieval, and purely visual tasks. These gains hold regardless of whether the model is trained on original web data or rich re-captioned data, demonstrating SuperCLIP's ability to recover textual supervision in both cases. Furthermore, SuperCLIP alleviates CLIP's small-batch performance drop through classification-based supervision that avoids reliance on large batch sizes.

## 1 Introduction

CLIP [45] has become a cornerstone in vision-language learning, excelling in tasks like zero-shot classification, retrieval, and text-to-image generation [72, 29, 26, 46]. By aligning images and text in a shared embedding space and leveraging large-scale noisy web data [11, 6, 47], it learns rich, transferable representations. However, despite its strong performance, CLIP's representations still have room for improvement, and further enhancing them remains crucial for advancing multimodal applications [51, 65, 52, 57].

Recent works have proposed various improvements to CLIP in three main dimensions: training strategies [33, 67, 42, 32, 22, 64, 12, 60], architectural modifications [5, 56, 63, 70, 16, 58, 50], and data collection techniques [23, 17, 27, 38, 1, 11, 59]. These approaches have significantly enhanced the performance of the CLIP model in zero-shot and other downstream tasks [55].

However, despite these advances, an interesting phenomenon emerges: Contrastive models like CLIP still struggle to fully exploit the rich supervision in captions, especially when those captions are long and detailed re-captioned texts [31, 10, 27]. This counterintuitive phenomenon highlights a fundamental issue: contrastive learning fails to make full use of fine-grained semantic signals in text, even when they are explicitly available.

---

*Corresponding author (zilong.huang2020@gmail.com).

39th Conference on Neural Information Processing Systems (NeurIPS 2025).

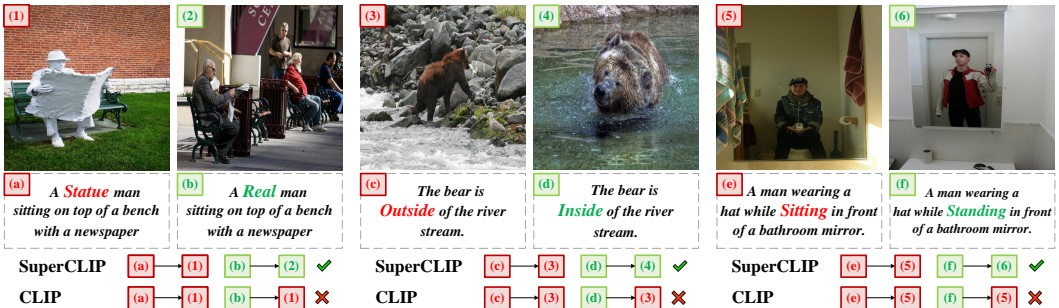

Figure 1: **Evaluating Fine-Grained Alignment in Image-Text Retrieval.** Each row presents pairs of images and captions that are visually and semantically very similar, but differ in fine-grained semantic distinctions such as object status (e.g. **Statue** vs. **Real**), spatial relation (e.g. **Outside** vs. **Inside**), and action (e.g. **Sitting** vs. **Standing**). While both images and texts are close in meaning, SuperCLIP demonstrates a stronger ability than CLIP in correctly distinguishing these fine-grained semantic distinctions. Additional examples are provided in **Appendix A.1.**

This challenge stems from how CLIP is trained by optimizing only for global image-text similarity, while overlooking the dense semantic cues encoded in individual words or phrases [70, 68, 61, 36, 24, 62]. The problem is further compounded by the characteristics of typical web data, which tend to be short, ambiguous, and only loosely aligned with the visual content [11]. As a result, CLIP-like models often miss subtle but important distinctions in object attributes, spatial relationships, and actions. As illustrated in Fig.1, CLIP may confuse a statue with a real person or fail to distinguish whether a bear is inside or outside the river. This lack of fine-grained alignment limits their effectiveness in downstream multimodal tasks that require precise visual-textual understanding [54]. Existing works have attempted to address this issue, but they either rely on additional annotated datasets beyond the web-scale data typically used for CLIP training, or introduce substantial computational overhead [31, 70, 68, 30, 64, 24, 62, 54, 55].

Thus, in this work, we propose SuperCLIP, a super simple yet effective approach that introduces a classification-based supervision method [20] into the contrastive learning paradigm of image–text pretraining. With only a lightweight linear layer added to the vision encoder, SuperCLIP directly leverages raw text tokens to guide the vision encoder to attend to semantic entities mentioned in the text and their visual manifestations in the image. In this way, SuperCLIP fully leverages the rich textual supervision from all words in the text, thereby enhancing the model's ability to achieve fine-grained visual-text alignment — with just a 0.077% increase in total FLOPs, and without requiring additional annotated data.

Extensive experiments demonstrate that our method effectively helps CLIP models recover rich textual supervision from all words in the text—whether trained on original web data or rich re-captioned data—leading to consistent improvements in zero-shot performance on classification and retrieval tasks, while also enhancing the vision encoder's features for purely visual tasks. Furthermore, SuperCLIP is simple and easy to implement, making it readily applicable to other CLIP-style training frameworks such as SigLIP [67] and FLIP [33], where it also brings consistent performance gains. Finally, thanks to its classification-based supervision and independence from large batch sizes, SuperCLIP alleviates the performance degradation typically observed in CLIP under small-batch training settings.

Our main contributions can be summarized as follows:

1. We propose SuperCLIP, a simple yet effective vision–language pretraining framework that seamlessly integrates classification-based supervision into contrastive learning, enabling CLIP models to effectively recover rich textual supervision from all words in the text.

2. Without introducing heavy computational cost or requiring additional annotated data, Super-CLIP enhances CLIP's ability to achieve fine-grained visual-text alignment and mitigates its performance degradation under small batch sizes.

3. Empirical results demonstrate that SuperCLIP achieves improved performance on zero-shot classification and retrieval tasks, as well as on purely visual downstream tasks, thereby confirming its broad effectiveness.

## 2 Related Work

In this section, we first provide a brief overview of representative efforts to improve CLIP. Then, we discuss existing approaches that specifically aim to address the underlying limitations of CLIP highlighted in the introduction.

### 2.1 Contrastive Vision-Language Pretraining

Contrastive learning has become the dominant approach for vision-language pretraining, with CLIP [45] demonstrating strong zero-shot transfer by aligning images and texts in a shared embedding space using large-scale noisy web data. Subsequent efforts have improved CLIP along three major dimensions. **Training-centric** [33, 67, 42, 32, 22, 64, 12, 60] strategies improve learning efficiency and robustness by modifying optimization objectives and training dynamics, such as using a sigmoid-based contrastive loss in SigLIP [67], applying masked image modeling to accelerate training in FLIP [33], and leveraging nearest-neighbor supervision to enhance data efficiency in DeCLIP [32]. **Model-centric** [5, 56, 63, 70, 16, 58, 50] improvements include designing stronger vision encoders such as Vitamin [5], rethinking input representations as in CLIPPO [56], and introducing more robust attention mechanisms like DiffCLIP [16]. **Data-centric** [23, 17, 27, 38, 1, 11, 59]. approaches focus on scaling dataset size and diversity to enhance model generalization, exemplified by ALIGN [23], LAION-5B [47], and DataComp [11]. In summary, these methods have effectively boosted the CLIP model's performance by refining the data, model architecture, and training techniques.

### 2.2 Improve CLIP with Additional Supervision

A number of recent works have considered the underlying problem that CLIP struggles with fine-grained visual-text alignment due to its reliance on global image-text similarity and weak, ambiguous supervision from web-sourced captions [31, 70, 68, 30, 64, 24, 62, 54, 55]. To address this issue, these works introduce additional forms of supervision to enhance fine-grained visual-text alignment. Recap-DataComp-1B [31] recaptions the original web data using LLaMA-3 to produce more informative captions for improving CLIP, but their findings show that CLIP is not fully effective at leveraging such rich textual supervision. While RegionCLIP [70] introduces region-level supervision without manual labels, it inherits CLIP's semantic limitations, overlooks inter-region relationships, and incurs additional computation due to region proposal processing. Long-CLIP [68] extends CLIP to long-text understanding via positional embedding stretching and component matching, but it compromises zero-shot image classification performance by disrupting the alignment between short-text prompts and visual features. UniCL [64] enhances CLIP by unifying contrastive learning across image-text and image-label pairs, but it relies on additional human-annotated category labels, which limits its scalability compared to purely web-supervised approaches. Eyes Wide Shut [54] and SigLIP 2 [55] both improve visual grounding and understanding through dense feature integration, but their methods introduce substantial computational and data overhead. In summary, these methods either fall short of fully resolving the issue, rely on additional annotated datasets beyond the web-scale data typically used for CLIP training, or introduce substantial computational overhead.

## 3 Motivation and Method

In this section, we first revisit the inherent limitations of the CLIP contrastive learning paradigm to motivate our approach. Then, we present a super simple classification-based method to recover rich textual supervision and improve CLIP's fine-grained visual-text alignment. The overall framework of our proposed SuperCLIP is illustrated in Fig.2.

### 3.1 Limitations of the Contrastive Learning Paradigm

**Overall Review of CLIP Training**  CLIP [45] learns joint image-text embeddings using a large collection of paired examples $\{(I_k, T_k)\}_{k=1}^{M}$. The model consists of two encoders—$f_\theta$ for images

| Keyword Group | Man + Newspaper | | Bear + River | | Man + Mirror | |
|---|---|---|---|---|---|---|
| Condition | Basic Pair | + Real/Stat | Basic Pair | + In/Out | Basic Pair | + Sit/Stand |
| Matching Captions | 333 | 6 | 219 | 2 | 1216 | 19 |
| Percentage (%) | 0.00333 | 0.0006 | 0.00219 | 0.00002 | 0.01216 | 0.00019 |

Table 1: **Keyword Co-occurrence Statistics in Datacomp-1B** [11] **(10M captions).** "In/Out" = Inside/Outside; "Real/Stat" = Real/Statue; "Sit/Stand" = Sitting/Standing. Each column shows how many captions match specific keyword combinations. Percentages refer to frequency in 10M captions. More keyword combination results are provided in **Appendix A.2.**

and $g_\phi$ for text—and normalizes their outputs to unit length. Specifically, for each image $I_i$ and text $T_i$, their embeddings are computed as:

$$u_i = \frac{f_\theta(I_i)}{\|f_\theta(I_i)\|_2}, v_i = \frac{g_\phi(T_i)}{\|g_\phi(T_i)\|_2}. \tag{1}$$

For a batch of $N$ pairs, CLIP computes the similarity matrix:

$$S_{ij} = \frac{u_i^\top v_j}{\tau}, \tag{2}$$

where $\tau$ is the temperature parameter. The objective function $\mathcal{L}_{\text{CLIP}}$ is designed to maximize the similarity between matching image-text pairs while minimizing the similarity between non-matching pairs in the shared embedding space. It is defined as:

$$\mathcal{L}_{\text{CLIP}} = -\frac{1}{2N} \sum_{i=1}^{N} \left( \log \frac{\exp(S_{ii})}{\sum_{k=1}^{N} \exp(S_{ik})} + \log \frac{\exp(S_{ii})}{\sum_{k=1}^{N} \exp(S_{ki})} \right), \tag{3}$$

where $\log \frac{\exp(S_{ii})}{\sum_{k=1}^{N} \exp(S_{ik})}$ corresponds to the image-to-text part, and $\log \frac{\exp(S_{ii})}{\sum_{k=1}^{N} \exp(S_{ki})}$ corresponds to the text-to-image part. By aligning matching image-text pairs and separating non-matching ones, CLIP learns robust visual and textual features, which are applicable to various downstream tasks.

**Impact of Batch Size and Web Data Sparsity**   A key assumption of CLIP is that each batch must contain enough positive and negative pairs for effective learning [45]. When batch size is small, performance degrades rapidly [67], which is why CLIP training typically relies on very large batches—often 16k or more—demanding significant computational resources [33, 49]. Large batches help CLIP learn diverse object categories from web data [23, 47, 11], contributing to its strong zero-shot classification performance. Despite CLIP's strong performance in object recognition, it struggles with fine-grained attributes like actions, spatial relations, and object states. This is largely due to the nature of web data, where captions are often short, ambiguous, and poorly structured [11]. As a result, semantic combinations needed to learn fine-grained distinctions are rare and inconsistent. For example, "man + newspaper" appears 333 times in 10M captions, but "man + newspaper + real/statue" appears only 6 times, and some combinations—like "bear + river + in/out"—are nearly absent (see Table 1). These low-frequency cases rarely form effective contrastive pairs [2, 3, 54], and even when they do exist, they are unlikely to co-occur in the same batch—making contrastive learning of such concepts nearly impossible without extremely large, computationally expensive batch sizes.

**Limitation in Using Rich Textual Supervision**   CLIP is trained with a contrastive objective that aligns image and text embeddings by pulling matched pairs closer and pushing mismatched pairs apart within each batch. While effective at capturing global semantic alignment, this objective tends to overlook fine-grained or detailed semantic information present in the captions [70, 68, 61, 36, 24, 62], limiting the model's ability to fully leverage rich textual supervision. An interesting exploration in [31] re-captioned web data using powerful MLLMs like LLaMA-3 [14], enriching the captions with more semantic information. In theory, this should provide stronger supervision and improve performance. However, contrary to expectations, CLIP models trained under the contrastive learning paradigm actually showed a drop in performance when the original data was entirely replaced with re-captioned data. Our experiments in section 4.3 further confirm this phenomenon, showing that simply enriching captions does not necessarily lead to better performance under the contrastive learning paradigm. This suggests that CLIP's contrastive learning paradigm struggles to take advantage of rich textual supervision—in fact, the added complexity introduced by richer captions can hinder learning and lead to a drop in performance.

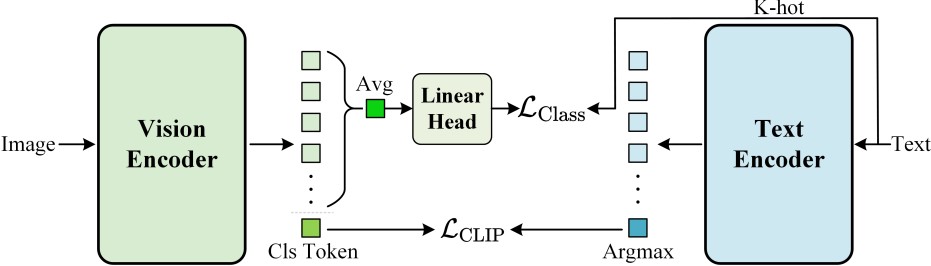

Figure 2: **Overall Framework of Our Proposed SuperCLIP.** Introducing simple classification-based supervision into the CLIP framework is straightforward. It only involves adding a lightweight linear layer to the image encoder to map the averaged image features to text classification targets, without requiring any changes to the original contrastive learning paradigm.

## 3.2 Super Simple Classification-based Supervision

Using text as supervision for visual backbones is well studied [41, 53, 25, 28]. However, existing methods often rely on manual filtering or heuristics to construct classification vocabularies [19, 40], which limits scalability to noisy web data. To overcome this, we follow [20] and use raw text tokens—prior to CLIP's text encoder—as direct classification labels for the vision encoder. Specifically, consider an image-text dataset $\mathcal{D} = \{(I_i, T_i) \mid i \in [1, N]\}$ used for contrastive training following the CLIP framework. Each caption $T_i$ is tokenized using CLIP's subword-level tokenizer with a vocabulary size of $V$, resulting in a set of token IDs $\mathcal{C}$. These tokens are then represented as a $V$-dimensional K-hot vector $\mathbf{y} \in \mathbb{R}^V$, where $y_c = 1$ if $c \in \mathcal{C}$, and $y_c = 0$ otherwise. While the original label $\mathbf{y}$ treats all subwords equally, some frequent stopwords or generic terms carry less discriminative information. To address this imbalance, an Inverse Document Frequency (IDF) based weighting is applied:

$$w_c = \log\left(\frac{|\mathcal{D}|}{1 + \mathrm{df}(c)}\right), \tag{4}$$

where $|\mathcal{D}|$ denotes the total number of image-text pairs in the dataset, and $\mathrm{df}(c)$ is the document frequency of subword $c$, i.e., the number of captions in which $c$ appears. Using these weights, a normalized weighted label distribution $\hat{y}_c$ is computed as:

$$\hat{y}_c = \frac{w_c y_c}{\sum_{c'=1}^{V} w_{c'} y_{c'}}. \tag{5}$$

Given the normalized label distribution $\hat{y}_c$, the goal is to train the model such that its output distribution aligns closely with this weighted supervision signal. Let $x_c$ denote the logit output of the model for class $c \in \{1, \ldots, V\}$, obtained by applying a linear classification head on top of the image features extracted by the CLIP vision encoder. The final classification loss is defined as the cross-entropy between the weighted label distribution $\hat{y}_c$ and the softmax-normalized model predictions:

$$\mathcal{L}_{\mathrm{Class}} = -\sum_{c=1}^{V} \hat{y}_c \log\left(\frac{e^{x_c}}{\sum_{c'=1}^{V} e^{x_{c'}}}\right). \tag{6}$$

This classification loss encourages alignment between the model predictions and all subword tokens extracted from the text, ensuring that the full textual supervision signal is utilized. Finally, since both the training data and the vision encoder are taken directly from the existing CLIP training pipline (except for a simple linear layer that maps image features to classification targets), this loss can be easily added to the CLIP optimization objective:

$$\mathcal{L}_{\mathrm{Total}} = \mathcal{L}_{\mathrm{CLIP}} + \mathcal{L}_{\mathrm{Class}}. \tag{7}$$

In this way, our method extends CLIP to effectively recover rich textual supervision from all words in the text, naturally guiding the model to attend to fine-grained visual-text alignment that is often overlooked by standard CLIP. What's more, since the classification loss does not rely on batch size, it can alleviates the performance degradation typically observed in CLIP under small-batch training settings.

| Model | Pretrain | Image Classification | | Image Retrieval | | Text Retrieval | |
|---|---|---|---|---|---|---|---|
| | | val | v2 | COCO | Flickr | COCO | Flickr |
| CLIP | B-512M | 60.5 | 53.0 | 29.0 | 54.5 | 46.7 | 73.3 |
| SuperCLIP | B-512M | 63.5 (+3.0) | 55.2 (+2.2) | 31.3 (+2.3) | 56.9 (+2.4) | 47.8 (+1.1) | 75.6 (+2.3) |
| CLIP | L-512M | 66.1 | 57.4 | 32.7 | 57.0 | 49.6 | 76.4 |
| SuperCLIP | L-512M | 70.1 (+4.0) | 62.5 (+5.1) | 35.9 (+3.2) | 62.4 (+5.4) | 52.2 (+2.6) | 79.3 (+2.9) |
| CLIP | L-12.8B | 79.0 | 72.0 | 43.9 | 72.7 | 62.5 | 87.0 |
| SuperCLIP | L-12.8B | 80.0 (+1.0) | 72.8 (+0.8) | 45.5 (+1.6) | 74.2 (+1.5) | 63.1 (+0.6) | 88.1 (+1.1) |

Table 2: **Comparison with CLIP across Different Model Sizes.** We report **zero-shot** image classification accuracy (%) on ImageNet-1K (val and v2), and **zero-shot** image and text retrieval (Recall@1, %) on COCO and Flickr30K, comparing CLIP and our SuperCLIP under three settings: B-512M, L-512M, and L-12.8B, where models are pretrained on 512M or 12.8B samples from DataComp-1B. Values in parentheses reflect absolute gains or drops for SuperCLIP relative to CLIP.

## 4 Empirical Results

### 4.1 Experimental Setup

**Pretraining Setup.** We pretrain our proposed SuperCLIP and CLIP on a standard subset of the Datacomp dataset [11], which contains about 1.3B image-text pairs. All images are resized to a fixed resolution of $224 \times 224$, and the text is minimally processed with only basic tokenization. Note that, except for the experiment (**Comparison with CLIP with Mixed Caption**) in Section 4.3 which consider useing the Recap-DataComp[31] data, all other experiments are conducted on the original Datacomp dataset. All experiments are conducted with a batch size of 16k, except for those under varying batch sizes analyzing the impact on CLIP. For fair comparison, all models adopt AdamW with a cosine schedule, using the same learning rate and weight decay as CLIP.

**Evaluation Protocol.** For zero-shot evaluation, we use the open-source LAION CLIP Benchmark framework [47] to assess all models on zero-shot classification and image-text retrieval. For linear probing image classification experiments, we follow the training protocol introduced in MAE [18]. For semantic segmentation and depth estimation, we follow a protocol similar to DINOv2 [44].

### 4.2 Main Results

**Comparison across Different Model Sizes.** We demonstrate that our method consistently benefits CLIP across different model sizes, through **zero-shot** image classification on ImageNet-1K [8] (val and v2) and image-text retrieval on COCO [35] and Flickr30K [66]. Detailed results are presented in Table 2. By training both B- and L-sized models with varying amounts of pretraining data, we compare CLIP and our proposed SuperCLIP under three settings: B-512M, L-512M, and L-12.8B (ViT-B/L pretrianed with seen 512M/12.8B samples). Under the B-512M setting, SuperCLIP improves classification and retrieval performance across all tasks, including gains of over +3% in classification accuracy and up to +2.4% in retrieval. With the L-512M setting, the improvements are more substantial, reaching up to +5.4% in image retrieval and over +5.1% in classification. At the largest scale (L-12.8B), SuperCLIP still delivers consistent improvements across all benchmarks. For the L-size model, we estimate the additional computation introduced by the linear head added for classification-based supervision (see Table 3) using a single image-text pair, which accounts for only 0.077%. These results demonstrate that our method not only scales well with model and data size, but also consistently enhances CLIP's performance by better leveraging classification supervision—without introducing significant computational overhead. More FLOPs statistics of the models are provided in **Appendix A.3.**

**Analysis of Performance Gain.** We analyze word-image similarity statistics to demonstrate that, compared to CLIP, our SuperCLIP more effectively captures fine-grained visual attributes beyond global semantics. Visualization and statistical results are presented in Fig.3 and Table 4. We compute the similarity between each image and the words in its captions on the COCO validation set, measuring how much attention the model gives to different words. For each word, we then average its similarity across the dataset by dividing the total similarity by its frequency. After filtering

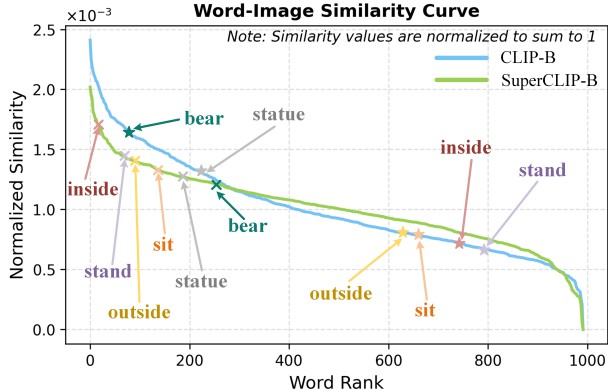

| Component | CLIP | SuperCLIP |
|---|---|---|
| Vision Encoder | 59.689 | 59.689 |
| Text Encoder | 6.547 | 6.547 |
| Linear Head | - | 0.051 |

Table 3: **FLOPs Count (GFLOPs).**

| Metric | CLIP | SuperCLIP |
|---|---|---|
| Total words | 992 | 992 |
| Std Deviation | 0.0340 | 0.0213 |
| Value Range | 0.2065 | 0.1401 |
| Mean Slope | 0.000208 | 0.000141 |
| Top-1→100 | 0.0702 | 0.0439 |

Figure 3: **Visualization of Word-Image Similarity Distribution.** We ranked the similarity scores of 1,000 words that appeared in the captions and highlighted the positions of fine-grained attributes discussed in the above Fig.1.

Table 4: **Statistical Summary.** Mean Slope ($\Delta$sim): Average drop in similarity between words as the rank goes down. Top-1→100: Difference in similarity between the 1st and 100th word.

out low-frequency words (fewer than 20 occurrences), we obtain a set of about 1,000 words and rank them by average similarity. The results are both interesting and expected. CLIP tends to rank object category words (e.g., zebras, kites, elephants, often in the top 20) highest, as they are easily captured by global visual features. In contrast, SuperCLIP, with added classification supervision, raises the ranks of words describing object status, spatial relations, and actions (see Fig. 3). This is because classification supervision encourages the model to focus more on fine-grained visual details overlooked by CLIP. As shown in Table 4, SuperCLIP also produces more stable similarity rankings with lower variance across words, reducing the long-tail effect seen in CLIP. Overall, these results show that SuperCLIP better captures fine-grained attributes that CLIP often overlooks, leading to improved performance across multiple tasks. More statistical analyses on word-image similarity are provided in **Appendix A.4.**

## 4.3 Recover Textual Supervision

**Comparison with CLIP using Mixed Caption.** We demonstrate that our method helps CLIP recover overlooked textual supervision through extensive experiments, including **zero-shot** classification on 38 datasets [47] and image-text retrieval on COCO and Flickr30k (Table 5). Following [31], we train our models on a mix of DataComp-1B and Recap-DataComp-1B, with the latter containing longer, semantically richer captions. While such captions provide more detailed and fine-grained supervision, increasing their proportion in training tends to degrade CLIP's performance—suggesting that contrastive learning alone may not fully benefit from rich textual signals. In contrast, the classification supervision introduced in our method is better equipped to utilize this additional semantic information, thereby mitigating the limitations of contrastive learning. Under the **DualCaption** setting, contrastive learning captures coarse-grained semantics from short captions, while our classifier extracts fine-grained details from long ones—achieving strong performance without the need for the carefully tuned **"0.8/0.2"** mixing ratio identified through extensive search in [31]. Complete evaluation results across 38 datasets are provided in **Appendix A.5.**

## 4.4 Generalization Analysis

**Generalize to Other CLIP-style Frameworks.** We test the generalizability of our method on two CLIP-style models, SigLIP and FLIP, using **zero-shot** classification on ImageNet-1K and image-text retrieval on COCO and Flickr30K. Detailed results are presented in Table 6. Our SuperCLIP variants (SuperSigLIP and SuperFLIP) consistently outperform their baselines under the same pretraining setup. SuperSigLIP achieves gains of up to +3.7% in image classification and +2.9% in image/text retrieval. Similarly, SuperFLIP improves by +3.4% in classification, +2.6% in image retrieval, and up to +5.3% in text retrieval. This demonstrates that our method is not limited to CLIP, but is a generally effective enhancement to vision-language pretraining.

| Model-Size | Mixed Caption | Image Retrieval | | Text Retrieval | | Image Classification |
|---|---|---|---|---|---|---|
| | Short / Long | COCO | Flickr | COCO | Flickr | Average. 38 |
| CLIP-B | 1.0 / 0.0 | 29.0 | 54.4 | 46.7 | 73.7 | 43.4 |
| SuperCLIP-B | 1.0 / 0.0 | **31.3** | **57.6** | **47.8** | **75.6** | **44.5** (+1.1) |
| CLIP-B | 0.0 / 1.0 | 23.6 | 41.8 | 40.5 | 66.2 | 27.8 |
| SuperCLIP-B | 0.0 / 1.0 | **30.6** | **48.7** | **47.2** | **70.4** | **31.4** (+3.6) |
| CLIP-B | 0.8 / 0.2 | 32.7 | 57.5 | 50.2 | 76.0 | 42.8 |
| SuperCLIP-B | Dual | **34.1** | **60.2** | **51.2** | **76.6** | **45.1** (+2.3) |
| CLIP-L | 1.0 / 0.0 | 32.7 | 57 | 49.6 | 76.4 | 45.7 |
| SuperCLIP-L | 1.0 / 0.0 | **35.9** | **62.4** | **52.2** | **79.3** | **48.6** (+2.9) |
| CLIP-L | 0.0 / 1.0 | 26.2 | 43.1 | 42.9 | 65.9 | 30.0 |
| SuperCLIP-L | 0.0 / 1.0 | **34.2** | **55.7** | **52.1** | **75.0** | **33.8** (+3.8) |
| CLIP-L | 0.8 / 0.2 | 37.0 | 61.1 | 53.7 | 78.8 | 46.8 |
| SuperCLIP-L | Dual | **37.6** | **65.3** | **54.0** | **82.5** | **49.5** (+2.7) |

Table 5: **Comparison with CLIP using Mixed Captions.** "Mixed Caption" refers to the ratio of short (DataComp-1B) and long (Recap-DataComp-1B) captions used during training. The **"0.8/0.2"** mix is the optimal ratio identified in [31] through extensive tuning. **"Dual"** denotes our setup where the contrastive loss uses only short captions and the classification loss uses only long captions. We report average **zero-shot** image classification accuracy (%) across 38 datasets, and **zero-shot** image/text retrieval (Recall@1, %) on COCO and Flickr30K, using **512M** training samples. **Bold** numbers indicate the best results, while values in parentheses show absolute gains or drops of SuperCLIP relative to CLIP.

| Model | Image Classification | | Image Retrieval | | Text Retrieval | |
|---|---|---|---|---|---|---|
| | val | v2 | COCO | Flickr | COCO | Flickr |
| SigLIP | 60.4 | 52.8 | 29.8 | 53.9 | 45.8 | 73.2 |
| SuperSigLIP | 64.1 (+3.7) | 55.9 (+3.1) | 32.5 (+2.7) | 56.8 (+2.9) | 48.6 (+2.8) | 75.9 (+2.7) |
| FLIP | 58.1 | 50.1 | 27.5 | 51.8 | 44.1 | 66.7 |
| SuperFLIP | 61.3 (+3.2) | 53.5 (+3.4) | 30.1 (+2.6) | 54.0 (+2.2) | 46.7 (+2.6) | 72.0 (+5.3) |

Table 6: **Generalization to Other CLIP-Style Frameworks.** We report **zero-shot** performance on image classification accuracy (%) on ImageNet-1K (val and v2), and image/text retrieval (Recall@1, %) on COCO and Flickr30K, comparing SigLIP and FLIP with their SuperCLIP variants (SuperSigLIP and SuperFLIP). All models are pretrained with 512M samples (B-512M). Numbers in parentheses indicate absolute gains over the original models.

**Enhance CLIP for Purely Visual Tasks.** We demonstrate how our method enhances CLIP for purely visual tasks, through **linear probing** image classification on ImageNet, semantic segmentation on Pascal [9] and ADE20K [71], and depth estimation on NYUv2 [43]. Detailed results are presented in Table 7. For linear probing image classification experiments, we freeze the backbone and train a linear classification head. For the semantic segmentation and depth estimation tasks, we similarly attach a linear head to the backbone, but fine-tune the entire model. SuperCLIP consistently improves performance across all tasks, indicating that the vision encoder trained with our method learns more effective and discriminative visual representations.

## 4.5 Impact of Batch Size

**Mitigate CLIP's Drop with Limited Batch Sizes.** We examine the extent to which our method mitigates CLIP's performance degradation under small batch sizes, through **zero-shot** and **linear probing** classification on ImageNet across batch sizes ranging from 1K to 32K. Detailed results are presented in Fig.4. For zero-shot classification (Left), SuperCLIP shows clear advantages under small-batch training, where CLIP suffers significant degradation. For linear probing (Right), SuperCLIP

| Model | Pretrian | Class ↑ | Segmentation ↑ | | Depth ↓ |
|---|---|---|---|---|---|
| | | ImageNet-1K | PASCAL | ADE20k | NYUv2 |
| CLIP | B-512M | 75.6 | 57.8 | 28.0 | 0.768 |
| SuperCLIP | B-512M | 77.1 (+1.5) | 65.5 (+7.7) | 32.1 (+4.1) | 0.746 (-0.022) |
| CLIP | L-512M | 79.7 | 67.8 | 34.2 | 0.740 |
| SuperCLIP | L-512M | 81.0 (+1.3) | 71.2 (+3.4) | 36.3 (+2.1) | 0.733 (-0.007) |

Table 7: **Enhance CLIP for Purely Visual Tasks.** We report performance on three purely visual tasks: **linear probing** image classification(Class) on ImageNet-1K (Accuracy, %), semantic segmentation(Segmentation) on PASCAL and ADE20K (mIoU), and depth estimation(Depth) on NYUv2 (RMSE). We compare CLIP and SuperCLIP under identical pretraining and evaluation settings to ensure a fair comparison across all purely visual tasks. Numbers in parentheses indicate absolute improvements over the original CLIP models.

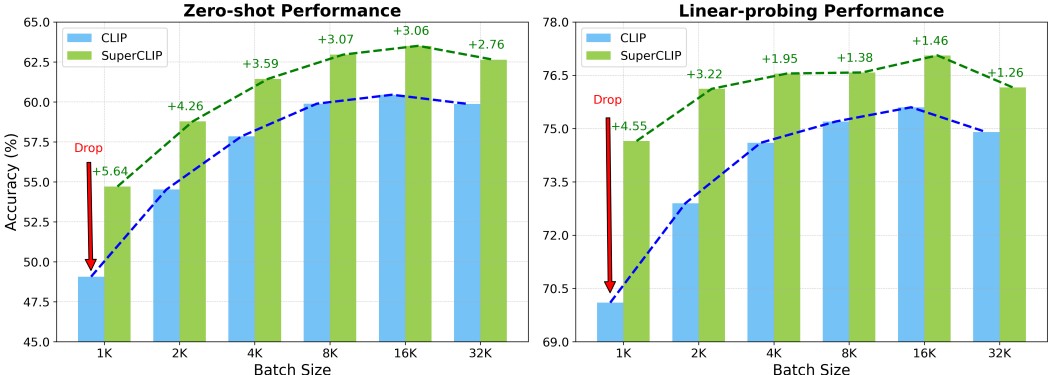

Figure 4: **Mitigate CLIP's Drop with Limited Batch Sizes.** We report zero-shot (Left) and linear-probing (Right) image classification accuracy (%) on ImageNet-1K (val) under varying batch sizes. The green bars represent the performance of SuperCLIP under different batch sizes, while the gray bars indicate the performance of CLIP under the corresponding batch sizes. Green numbers indicate absolute improvements over the original CLIP models at the corresponding batch sizes.

maintains stable performance across all batch sizes, preserving high-quality visual representations even in low-resource settings where CLIP's performance drops noticeably. The above results validate that our method effectively mitigates the performance degradation of CLIP under limited batch size conditions. This improvement is attributed to the introduction of classification supervision, which is inherently insensitive to batch size.

## 4.6 Integrate in Multi-modal LLM

**Compare with CLIP under Multi-modal LLM Setting.** We evaluate SuperCLIP beyond CLIP-style contrastive pretraining by integrating both CLIP and SuperCLIP (ViT-B/16, 512M samples) into the LLaVA-1.5 framework [37], combined with the Vicuna-7B language model [7], enabling a fair comparison within the multi-modal LLM setting. This setup supports effective multi-modal reasoning and instruction following across a broad range of downstream benchmarks, including VQAv2 [13], GQA [21], VizWiz [15], T-VQA [48], SQA [69], MMB (MMBench) [39], MME [4] and POPE [34]. Detailed results are presented in Table 8. These experiments confirm SuperCLIP's superior performance over CLIP encoders across multiple benchmarks, particularly on VQAv2 and MMBench, which focus on general visual question answering and fine-grained recognition, respectively. The strong transfer performance demonstrates that SuperCLIP is not only effective in contrastive pretraining but also exhibits excellent cross-modal generalization when integrated into large-scale multi-modal frameworks.

## 4.7 Additional Ablation Studies

**Ablation on Loss Weighting and IDF Weighting** We study the effect of weighting the classification loss by multiplying the $\mathcal{L}_{\text{Class}}$ term in Eq. 7 by a factor $\lambda$. As shown in Table 9, performance improves

| Model | Pretrian | Vision & Language Downstream Tasks | | | | | | | |
|---|---|---|---|---|---|---|---|---|---|
| | | VQAv2 | GQA | VizWiz | T-VQA | SQA | MMB | MME | POPE |
| CLIP | B-512M | 67.8 | 55.4 | 42.1 | 47.8 | **69.3** | 49.1 | 1453 | 81.7 |
| SuperCLIP | B-512M | **69.6** | **57.5** | **44.4** | **48.4** | 69.1 | **55.9** | **1562** | **82.0** |

Table 8: **Compare with CLIP under Multi-modal LLM Setting.** We report the performance scores on 8 vision & language downstream tasks. **Bold** numbers indicate the best result.

| Task | Weighting Factor ($\lambda$) | | | | |
|---|---|---|---|---|---|
| | **0.4** | **0.6** | **1** | **1.4** | **1.6** |
| Classification | 44.1 | 45.0 | 47.1 | 46.9 | 47.2 |
| Image Retrieval | 41.3 | 42.1 | 44.0 | 43.8 | 44.2 |
| Text Retrieval | 58.3 | 59.8 | 61.0 | 60.9 | 62.0 |

Table 9: **Loss Weighting.** We report **zero-shot** classification accuracy (%) on ImageNet-1K (val) and the average retrieval result (Recall@1, %) across COCO and Flickr30K.

| Design | Image Retrieval | | Text Retrieval | | Classification |
|---|---|---|---|---|---|
| | **COCO** | **Flickr** | **COCO** | **Flickr** | **ImageNet-1K** |
| w/o IDF | 31.6 | 51.7 | 48.0 | 71.1 | 44.8 |
| IDF | **33.2** | **54.7** | **48.9** | **73.1** | **47.1** |

Table 10: **IDF Weighting.** We report **zero-shot** classification accuracy (%) on ImageNet-1K (val) and retrieval results (Recall@1, %) on COCO and Flickr30K, respectively.

across all tasks as $\lambda$ increases from 0.4 to 1.0. Beyond 1.0, text retrieval continues to improve, whereas image retrieval and classification saturate. This confirms the effectiveness of the classification loss, and we recommend $\lambda \geqslant 1.0$ in practice. Then, we assess the role of IDF weighting by comparing IDF-weighted and unweighted K-hot labels. As shown in Table 10, IDF consistently improves performance across all benchmarks, confirming its benefit. All additional ablation studies use a ViT-B/16 model trained on 256M samples with all other settings unchanged.

## 5   Conclusion and Future Work

We introduce SuperCLIP, a simple yet effective framework that adds classification supervision to CLIP-style vision–language pretraining. By treating raw text tokens as classification labels, SuperCLIP recovers rich semantic signals often missed by contrastive learning, enabling better use of full textual content beyond coarse image-text alignment. SuperCLIP consistently improves performance across a wide range of tasks, including zero-shot classification, image-text retrieval, linear probing, and purely visual benchmarks. It enhances CLIP's ability to achieve fine-grained visual-text alignment, while requiring no additional annotations or significant computational cost. Its batch-size-independent classification loss also mitigates CLIP's performance drop under small-batch settings, making it more practical for real-world applications. We hope these findings encourage further research into combining classification and contrastive learning in large-scale multimodal models. For the future work, SuperCLIP focuses on enhancing the supervision from text to the vision encoder. A promising direction is to explore whether a similar approach can improve the supervision from images to the text encoder.

**Acknowledgement**: This work was partially supported by the National Natural Science Foundation of China (No. 62276108).

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
