# OpenReview forum: "SuperCLIP: CLIP with Simple Classification Supervision"
_NeurIPS.cc/2025/Conference — NeurIPS 2025 poster_

### Official Review · Reviewer_2yeS · 2025-06-23

**Clarity:** 3
**Significance:** 3
**Originality:** 3
**Rating:** 4
**Confidence:** 4

**Summary:**

The problem of training CLIP with explicit classification supervision is considered. This paper addresses CLIP's limitation in utilizing fine-grained semantic signals in text and its performance drop with small batch sizes. By adding a lightweight linear layer to the vision encoder, SuperCLIP leverages raw text tokens as classification labels, improving fine-grained visual-text alignment with minimal computational overhead. Experiments show that SuperCLIP consistently improves zero-shot classification, image-text retrieval, and purely visual tasks, while also mitigating performance degradation under small batch sizes.

**Questions:**

Typos
L160 “Consider” -> ”consider”

Comments
1. In (7), did you try and tune any multiplier hyperparameters for balancing two loss terms?

**Ethical Concerns:**

["NO or VERY MINOR ethics concerns only"]

**Final Justification:**

Thank the authors for the response and addressing my concerns. I decided to maintain the rating of borderline accept.

**Limitations:**

yes

**Quality:**

3

**Strengths And Weaknesses:**

Strengthens
In addition to the standard info-NCE/contrastive losses, this work introduces another multi-label classification loss, which is based on the standard cross-entropy loss, but the target distribution is no longer one-hot. Different classnames were assigned with a different weights based on its tf-idf. The resulted CLIP was shown with better performance on 0-shot image classification and text-image retrieval.

Weaknesses
The issue of poor fine-grained alignment of CLIP is well-known in the community, and motivates the introduction of the topic of vision-language compositionality. A few examples are not sufficient to validate the claim that superCLIP is superior. More quantitative evaluations on relevant benchmarks for VL compositionality (e.g., SugarCrepe [T1], ARO [T2], VL-Checklist [T3], and more proposed after 2023).

[T1] SugarCrepe: A benchmark for faithful vision-language compositionality evaluation. NIPS’23
[T2] When and why vision-language models behave like bags-of-words, and what to do about it? ICLR’23.
[T3] Vl-checklist: Evaluating pre-trained vision-language models with objects, attributes and relations.

---

> ### Author Rebuttal · Authors · 2025-07-30
>
> > **Q1&W1.** More evaluations on benchmarks for VL compositionality.
>
> **A:** Thank you for the valuable suggestion. Following your advice, we have conducted extensive quantitative evaluations on several established benchmarks for vision-language compositionality, including SugarCrepe [T1], ARO [T2], and VL-Checklist [T3].
>
> Furthermore, to strengthen our validation, we have included results on Winoground [T4], a standard benchmark for visio-linguistic compositional reasoning, and the recently proposed ColorSwap [T5] benchmark (introduced after 2023), which specifically targets color-object compositional understanding through word order variation.
>
> To ensure a fair comparison, both the CLIP and SuperCLIP models used in this evaluation were trained under identical settings: a batch size of 16k, 512M seen samples, and ViT-L/16 as the backbone. The experimental results are as follows:
>
> **SugarCrepe**
>
> | Model      | Average |             |     REPLACE     |         | SWAP             |         |      ADD     |             |         |
> |------------|---------|--------------------|---------|---------|------------------|---------|---------|------------------|---------|
> |       -     |     -    | Object             | Attribute | Relation | Object           | Attribute | Object           | Attribute |
> | CLIP       | 73.7    | 92.4               | 79.5    | 63.7    | 60.8             | 60.8    | 81.2             | **77.4**  |
> | SuperCLIP  | **75.1**| **93.6**           | **81.7**| **64.4**| **62.9**         | **62.6**| **83.6**         | 77.2     |
>
>
> **ARO**
> | Model        | Average | VG-Relation | VG-Attribution |
> |--------------|---------|-------------|----------------|
> | CLIP         | 51.1    | 45.4        | 56.7           |
> | SuperCLIP| **51.4** | **45.7**     | **57.1**        |
>
> **VL-Checklist**
>
> | Model      | Average | Object | Attribute | Relation |
> |------------|---------|--------|-----------|----------|
> | CLIP       | 61.9    | 74.4   | 63.8      | 47.5     |
> | SuperCLIP | **63.4** | **76.3** | **66.2**   | **47.6**   |
>
> **Winoground**
>
> | Model      | image_acc | text_acc | acc  |
> |------------|-----------|----------|------|
> | CLIP       | 6.5       | 25.0     | 5.0  |
> | SuperCLIP| **8.0**     | **26.5**   | **6.0** |
>
> **ColorSwap**
>
> | Model      | text_score | image_score | group_score |
> |------------|------------|-------------|-------------|
> | CLIP       | 36.0       | **16.3**    | 11.3        |
> | SuperCLIP| **39.3**    | 14.7        | **11.7**     |
>
> These experiments demonstrate that SuperCLIP consistently achieves superior performance compared to CLIP under the same training settings on nearly all benchmarks, validating its enhanced compositional capabilities. We will include the corresponding results and analysis in the revised manuscript.
>
> [T1] SugarCrepe: A benchmark for faithful vision-language compositionality evaluation. NIPS’23
>
> [T2] When and why vision-language models behave like bags-of-words, and what to do about it? ICLR’23.
>
> [T3] Vl-checklist: Evaluating pre-trained vision-language models with objects, attributes and relations.
>
> [T4] Winoground: Probing Vision and Language Models for Visio-Linguistic Compositionality. CVPR’22.
>
> [T5] ColorSwap: A Color and Word Order Dataset for Multimodal Evaluation. ACL'24
>
> > **Q2.** Typos.
>
> **A:**  Thank you for your attentive reading and for pointing out this detail. We have corrected the typo as you suggested.
>
> > **Q3.** Ablation on Loss Weighting.
>
> **A:**  Thank you for your insightful suggestion. We have conducted an ablation study on loss weights  λ to investigate their impact:
>
>  L_Total = L_CLIP + λ × L_Class.
>
> To expedite the experiments without affecting the main conclusions, we used a ViT-B/16 model trained on 256M seen samples (the default in the paper is 512M). All other settings remained consistent with those described in the paper.
>
> **(Experiments with λ = 0.8 and 1.2 is still ongoing due to time and resource constraints.)**
>
> | λ            | Imagenet val | COCO T2I | Flicker T2I | COCO I2T | Flicker I2T |
> |--------------|--------------|----------|-------------|----------|-------------|
> | 0.4          | 44.1         | 30.9     | 51.6        | 47.3     | 69.3        |
> | 0.6          | 45.0         | 31.7     | 52.4        | 47.8     | 71.7        |
> | 0.8 (ongoing)|              |          |             |          |             |
> | 1            | 47.1         | 33.2     | 54.7        | 48.9     | 73.1        |
> | 1.2 (ongoing)|              |          |             |          |             |
> | 1.4          | 46.9         | 32.6     | 54.9        | 48.5     | 73.2        |
> | 1.6          | 47.2         | 33.5     | 55.0        | 49.9     | 74.1        |
>
> As λ increases from 0.4 to 1.0, model performance consistently improves across all tasks. When λ is increased beyond 1.0, Text-Retrieval (I2T) tasks continue to improve, while Image-Retrieval (T2I) and classification performance begin to plateau. This demonstrates the effectiveness of introducing the classification loss, and we recommend using a weight of 1.0 or higher for L_Class in practice.

---

> > ### Comment · Reviewer_2yeS · 2025-08-05
> >
> > Thank the authors for the response and addressing my concerns. I decided to maintain the rating of borderline accept.

---

> > > ### Author Response · Authors · 2025-08-05
> > >
> > > Thank you for your feedback and for taking the time to review our work. We appreciate your recognition of our efforts and will continue to improve our research based on your suggestions.

---

> ### Author Response · Authors · 2025-08-05
>
> Dear Reviewer,
>
> Thank you for your valuable comments and questions. We have done our best to address them in our responses. Please kindly let us know if there are any remaining concerns or points that need further clarification.
>
> Sincerely,
>
> SuperCLIP Authors

---

### Official Review · Reviewer_jiRy · 2025-07-02

**Clarity:** 3
**Significance:** 2
**Originality:** 2
**Rating:** 4
**Confidence:** 4

**Summary:**

This paper presents SuperCLIP, an enhancement to the widely used CLIP vision-language model, aiming to better leverage fine-grained semantic information from textual captions during training. SuperCLIP introduces simple yet effective framework that augments contrastive learning with classification-based supervision by adding only a lightweight linear layer to the vision encoder.

**Questions:**

See above in Weaknesses.

**Ethical Concerns:**

["NO or VERY MINOR ethics concerns only"]

**Final Justification:**

The rebuttal satisfactorily addresses key concerns.

**Limitations:**

yes

**Quality:**

2

**Strengths And Weaknesses:**

### Strengths:

SuperCLIP consistently surpasses CLIP across multiple benchmarks (Tables 2,5-7), demonstrating superior performance in zero-shot classification, COCO/Flickr30K retrieval, and visual tasks including segmentation and depth estimation.

### Weaknesses:

1. In the abstract, "This stems from CLIP's training objective, which optimizes only global image-text similarity and overlooks token-level supervision—limiting its ability to achieve fine-grained visual-text alignment." I think you have noticed this phenomenon, but could the solutions to this problem be compared more? There aren't enough baselines in main paper. the optimization method can be like what you said, mapping the averaged image features. It can also be in many other ways. For example, the way of introducing multi-class tokens (https://arxiv.org/pdf/2203.02891).

2. In Table 5, why did superCLIP-B/L decrease by almost the same proportion (44.5 -> 31.4/48.6 -> 33.8) after changing the condition to Long caption? Does this mean that your approach does not essentially address the "contrastive learning alone may not fully benefit from rich textual signals" problem you mentioned.

3. The Dual setting shows minimal gains over Short in classification tasks (44.5->45.1),(48.6->49.5), with almost no improvement. A deeper investigation into why this occurs would strengthen the paper.

4. Since CLIP is a relatively older vision-language model (VLM), the applicability of SuperCLIP remains limited, it has only been optimized for CLIP. Could try the method on other VLM models.

---

> ### Author Rebuttal · Authors · 2025-07-31
>
> > **Q1** More Comparisons with Multi-Class Token Based or Other Alternatives
>
> **A:**  Thank you for the thoughtful suggestion.
>
> **(1) More Comparisons with Fine-Grained Alignment Baselines:**
>
> As discussed in the Introduction, many existing methods addressing CLIP’s fine-grained alignment either rely on extra labeled data or introduce significant complexity. In contrast, our method requires no additional supervision, is simple, and easily scalable to CLIP-style pretraining.
>
> That said, we acknowledge that some recent approaches—such as DetailCLIP [T1], A-CLIP [T2], and MaskCLIP [T3]—also focus on improving fine-grained alignment during pretraining only, without using extra data. Although their designs are more complex and computationally costly, comparison with them is still meaningful.
> To that end, we have added experiments following DetailCLIP’s setup (15M samples, batch size 4K, 25 epochs) and conducted zero-shot evaluation on standard benchmarks. The experimental results are as follows:
>
> | Model       | Flickr30K (I2T) | Flickr30K (T2I) | COCO (I2T) | COCO (T2I) | Imagenet-1k |
> |-------------|------------------|------------------|-------------|-------------|---------------|
> | SLIP [T4]    | 57.2             | 41.2             | 33.6        | 21.9        | 42.8          |
> | MaskCLIP     | 60.0             | 38.8             | 34.1        | 21.2        | 42.7          |
> | A-CLIP       | 62.7             | 42.1             | 38.0        | 23.2        | 43.9          |
> | DetailCLIP   | 62.8             | 42.2             | 38.3        | 22.9        | 43.9          |
> | SupeCLIP     | **65.8**         | **51.9**         | **43.2**    | **30.3**    | **44.8**      |
>
> As shown in the results, SupeCLIP achieves the best performance across all benchmarks, with particularly clear gains on retrieval tasks, while remaining significantly simpler. In contrast, DetailCLIP—though effective—relies on heavier mechanisms such as token filtering, self-distillation, and reconstruction losses. This demonstrates that SupeCLIP offers a simple yet highly effective solution for enhancing fine-grained visual-text alignment in CLIP-style models.
>
> **(2) On Multi-Class Token Approaches:**
>
>  While methods like the Multi-Class Token Transformer [T5] are effective in segmentation tasks, applying them to CLIP-style pretraining is nontrivial. Such methods require predefined class labels per image—unavailable in CLIP’s web-scale, caption-only training setup. Extending such methods to CLIP would require introducing ~50K unique class tokens—one per subword—and feeding them alongside the image into the ViT, resulting in prohibitive computational cost.
>
> Our approach instead reuses existing caption tokens via a simple classification head, avoiding external labels or overhead—making it a more scalable and compatible solution for CLIP.
>
> [T1] DetailCLIP: Detail-Oriented CLIP for Fine-Grained Tasks. ICLR’25
>
> [T2] Attentive Mask CLIP. ICCV’23.
>
> [T3] MaskCLIP: Masked Self-Distillation Advances Contrastive Language-Image Pretraining. CVPR’23
>
> [T4] SLIP: Self-supervision meets Language-Image Pre-training. ECCV’22
>
> [T5] Multi-class Token Transformer for Weakly Supervised Semantic Segmentation. CVPR’22
>
> > **Q2** Addressing Rich Text Utilization in Long Caption Scenarios
>
> **A:**  The performance drop when switching from short to long captions generated by LLMs/MLLMs is a common phenomenon. Prior works such as Long-CLIP [T1], VeCLIP [T2], and What-If [T3] have all reported similar trends. This degradation is likely due to the lack of data diversity in model-generated data (long captions) [T1, T2, T3].
>
> Furthermore, our experimental results indicate that SuperCLIP is better at leveraging long-caption data than the standard CLIP model. As shown in Table 5, while both CLIP and SuperCLIP experience performance drops with long captions, SuperCLIP-B/L consistently mitigates this degradation. Specifically, SuperCLIP-B/L outperforms CLIP-B/L by 1.1/2.9 points with short captions and extends this lead to a larger margin of 3.6/3.8 points with long captions, demonstrating its superior ability to handle longer, more descriptive text.
>
> Moreover, our Dual setting directly targets the issue raised by your question: it uses short captions for the contrastive branch and long captions for the classification branch, effectively combining their strengths. This strategy yields a 2.3/2.7-point improvement over the best-performing CLIP-B/L baseline trained on a carefully balanced caption mix.
>
> In summary, while contrastive learning alone struggles with rich textual signals, our method explicitly addresses and alleviates this limitation. We will make this clarification more explicit in the revised manuscript.
>
> [T1] Long-CLIP: Unlocking the Long-Text Capability of CLIP. ECCV’24
>
> [T2] VeCLIP: Improving CLIP Training via Visual-enriched Captions. ECCV’24
>
> [T3] What If We Recaption Billions of Web Images with LLaMA-3? ICML’25
>
> > **Q3** Investigation of Limited Classification Improvement
>
> **A:**  Thank you for this insightful comment.
>
> Your comment touches on an important and subtle point in CLIP-style training with long captions. Our proposed Dual setting is essentially an exploratory attempt to integrate long captions into CLIP-style training. While our formulation is new, the impact of long captions on CLIP-like models has been well studied in recent work such as Long-CLIP [T1], VeCLIP [T2], and What-If [T3]. These studies show that while integrating long captions with short captions significantly benefits retrieval tasks—due to richer semantic details—it often has a negative effect on zero-shot classification. This is because classification relies on concise, prompt-like inputs, and long captions can dilute class-specific signals.
>
> Our Dual setting directly addresses this issue: it retains short captions for the contrastive branch to preserve strong classification performance, while the classification branch leverages long captions to enhance retrieval. As a result, although the classification gains are modest (44.5→45.1; 48.6→49.5), we observe substantial improvements in retrieval (SuperCLIP-B/L gain 2.45/2.4 points in Recall@1).
>
> In summary, compared to other methods  [T1,T2,T3] that incorporate long captions, which tend to degrade zero-shot classification performance, our approach maximizes retrieval performance without hurting classification — and even improves it.
>
> We appreciate your suggestion and will incorporate a clearer discussion of this design rationale in our revised manuscript.
>
> [T1] Long-CLIP: Unlocking the Long-Text Capability of CLIP. ECCV’24
>
> [T2] VeCLIP: Improving CLIP Training via Visual-enriched Captions. ECCV’24
>
> [T3] What If We Recaption Billions of Web Images with LLaMA-3? ICML’25
>
> > **Q4** Applicability of SuperCLIP to Other Vision-Language Models
>
> **A:**  Thank you for your insightful and constructive comment.
>
> **First**, although SuperCLIP was originally designed for CLIP, our results (Table 6) show consistent gains on recent variants like SigLIP and FLIP in both classification and retrieval tasks. This confirms its effectiveness across modern CLIP-style models.
>
> **Second**, while evaluating SuperCLIP beyond the original CLIP is valuable, it’s important to note that CLIP-style models still underpin most modern VLM systems. Recent models like LLaVA [T1], MiniGPT-4 [T2], and SmolVLM [T3] all adopt CLIP-based encoders such as EVA-CLIP [T4], SigLIP [T5], and FLIP [T6], highlighting the continued importance of improving CLIP-style training approaches like SuperCLIP.
>
> To assess SuperCLIP beyond contrastive VLMs, we integrate both CLIP and SuperCLIP (ViT-B/16, 512M samples) into the LLaVA-1.5 framework with Vicuna-7B [T7], allowing a fair comparison in the MLLM setting:
>
> | Model      | VQAv2 | GQA  | VizWiz | TextVQA | ScienceQA | MMBench | MME       | POPE |
> |------------|-------|------|--------|---------|-----------|---------|-----------|------|
> | CLIP       | 67.8  | 55.4 | 42.1   | 47.8    | **69.3**   | 49.1    | 1175/**278** | 81.7 |
> | SuperCLIP  | **69.6**  | **57.5** | **44.4**   | **48.4**    | 69.1     | **55.9**   | **1292**/270 | **82.0** |
>
> These experiments confirm SuperCLIP’s superior performance compared to standard CLIP encoders across multiple downstream datasets, particularly in visual question answering and fine-grained recognition tasks. These results demonstrate SuperCLIP’s general applicability and effectiveness in broader VLM architectures, beyond traditional contrastive models.
>
> **Third**, while these results are promising, exploring SuperCLIP’s applicability to VLMs beyond the CLIP family (e.g., non-contrastive or generative VLMs) is beyond the scope of this paper. We believe this is a valuable direction for future work and plan to investigate it in more depth.
>
> [T1] LLaVA: Large Language and Vision Assistant. NIPS’23
>
> [T2] MiniGPT-4: Enhancing Vision-Language Understanding with Advanced Large Language Models. ICLR’24
>
> [T3] SmolVLM: Redefining small and efficient multimodal models.
>
> [T4] EVA-CLIP: Improved Training Techniques for CLIP at Scale.
>
> [T5] Sigmoid Loss for Language Image Pre-Training. ICCV’23
>
> [T6] Scaling Language-Image Pre-training via Masking. CVPR’23
>
> [T7] Vicuna: An Open-Source Chatbot Impressing GPT-4 with 90%* ChatGPT Quality.

---

### Official Review · Reviewer_Uu27 · 2025-07-02

**Clarity:** 3
**Significance:** 3
**Originality:** 2
**Rating:** 4
**Confidence:** 3

**Summary:**

This paper identifies a key limitation in CLIP and similar contrastive vision-language models: their struggle to capture fine-grained semantic details due to an objective function that only optimizes for global image-text similarity. To address this, the authors propose SuperCLIP, a simple yet highly effective modification. SuperCLIP augments the standard contrastive learning objective with an auxiliary classification loss. This is achieved by adding a single linear layer to the vision encoder, which is trained to predict the presence of text tokens from the paired caption. This token-level supervision, weighted by IDF to manage common words, encourages the vision encoder to learn features that correspond to more detailed semantic concepts mentioned in the text. The method is computationally lightweight, requires no additional annotated data, and is shown through extensive experiments to consistently improve performance on zero-shot classification, image-text retrieval, and even purely visual downstream tasks. The authors also demonstrate that SuperCLIP effectively utilizes richer, re-captioned data (where standard CLIP fails to benefit) and mitigates the performance degradation of CLIP in small-batch training scenarios.

**Questions:**

see weakness

**Ethical Concerns:**

["NO or VERY MINOR ethics concerns only"]

**Final Justification:**

I have carefully considered the paper, the authors' rebuttal. The authors have addressed my concerns. I decide to main my score borderline accept (4).

**Limitations:**

yes

**Quality:**

3

**Strengths And Weaknesses:**

Strengths:
1. Simplicity and Elegance: The core idea is exceptionally simple and elegant. By adding a single linear layer and a standard cross-entropy loss, the authors introduce a powerful supervisory signal that directly addresses a well-known weakness of contrastive learning. This simplicity, combined with its effectiveness, is a significant strength. The claimed computational overhead is negligible (0.077% FLOPs increase), making it a highly practical and scalable solution.
2. Comprehensive and Rigorous Evaluation: The paper is supported by a strong and thorough set of experiments that convincingly validate the authors' claims.
3. Addressing Practical Limitations of CLIP: The paper successfully demonstrates that SuperCLIP mitigates CLIP's well-known sensitivity to batch size (Fig. 4). This is a significant practical contribution, as it potentially lowers the barrier to entry for training high-quality vision-language models by reducing the dependency on massive computational resources.

Weaknesses:
1. Ablation on Loss Weighting: The total loss is presented as L_Total = L_CLIP + L_Class, which implies a fixed weighting of 1.0 for the new classification loss. It is common for multi-task objectives to be sensitive to the relative weighting of their component losses. The paper would be strengthened by a brief discussion or ablation study on how performance varies with different weights for L_Class. While the current results are already strong, understanding this sensitivity would be valuable for future practitioners.
2. One-Sided Supervision: The proposed supervision flows from the text (as labels) to the vision encoder. The authors acknowledge this as a direction for future work in the conclusion. However, it would be interesting to briefly discuss if this one-way supervision has any secondary effects (positive or negative) on the text encoder's representations. The current experiments focus on the vision encoder's quality and the joint embedding space, but an analysis of the text encoder in isolation could provide a more complete picture.
3. Role of IDF Weighting: The use of IDF weighting is intuitive and standard practice for handling frequent but uninformative words. An ablation that compares the final performance with and without IDF weighting (i.e., using simple unweighted K-hot vectors as labels) would quantify the importance of this specific design choice.

---

> ### Author Rebuttal · Authors · 2025-07-31
>
> > **Q1** Ablation on Loss Weighting.
>
> **A:** Thank you for the suggestion. To examine how performance varies with different weights for L_Class, we introduce a scalar λ in the total loss formulation:
>
> L_Total = L_CLIP + λ × L_Class.
>
> To expedite the experiments without affecting the main conclusions, we used a ViT-B/16 model trained on 256M seen samples (the default in the paper is 512M). All other settings remained consistent with those described in the paper.
>
> **(Experiments with λ = 0.8 and 1.2 is still ongoing due to time and resource constraints.)**
>
> | λ            | Imagenet val | COCO T2I | Flicker T2I | COCO I2T | Flicker I2T |
> |--------------|--------------|----------|-------------|----------|-------------|
> | 0.4          | 44.1         | 30.9     | 51.6        | 47.3     | 69.3        |
> | 0.6          | 45.0         | 31.7     | 52.4        | 47.8     | 71.7        |
> | 0.8 (ongoing)|              |          |             |          |             |
> | 1            | 47.1         | 33.2     | 54.7        | 48.9     | 73.1        |
> | 1.2 (ongoing)|              |          |             |          |             |
> | 1.4          | 46.9         | 32.6     | 54.9        | 48.5     | 73.2        |
> | 1.6          | 47.2         | 33.5     | 55.0        | 49.9     | 74.1        |
>
> As λ increases from 0.4 to 1.0, model performance consistently improves across all tasks. When λ is increased beyond 1.0, Text-Retrieval (I2T) tasks continue to improve, while Image-Retrieval (T2I) and classification performance begin to plateau. This demonstrates the effectiveness of introducing the classification loss, and we recommend using a weight of 1.0 or higher for L_Class in practice.
>
> > **Q2** One-Sided Supervision
>
> **A:** Thank you for the suggestion. To investigate this, we have conducted a focused analysis of the text encoder in isolation using the Massive Text Embedding Benchmark (MTEB) [T1]. This evaluation allows us to more fully understand whether the one-way supervision mechanism in SuperCLIP influences the quality of textual representations.
>
> As shown in the results below, we compare CLIP and SuperCLIP across 8 representative text embedding tasks, including semantic similarity (STSBenchmark, BIOSSES), text classification (AmazonCounter, Emotion), clustering (TwentyNews, Reddit), and retrieval (SciFact, Touche2020):
>
> | Model           | semantic similarity |                 | classification  |         | clustering   |         | retrieval   |           |
> |----------------|---------------------|-----------------|------------------|---------|--------------|---------|-------------|-----------|
> |                | STSBenchmark        | BIOSSES         | AmazonCounter    | Emotion | TwentyNews   | Reddit  | SciFact     | Touche2020|
> | CLIP-B/16      | 0.69                | **0.66**        | 0.59             | 0.25    | 0.42         | 0.49    | 0.28        | 0.11      |
> | SuperCLIP-B/16 | **0.71**            | 0.65            | **0.60**         | **0.26**| **0.43**     | **0.51**| **0.33**    | **0.14**  |
>
> From these results, we observe that the SuperCLIP text encoder consistently achieves slightly better performance than the original CLIP across almost all tasks. This indicates that, although our classification supervision is applied only to the vision encoder, the text encoder does not degrade—in fact, it appears to benefit indirectly from the more effective supervision introduced by the classification loss, which enhances the overall training dynamics of the joint model.
>
> However, we acknowledge that the improvement is modest and not the primary objective of SuperCLIP. In future work, we will further explore effective supervision flowing from images (as labels) to the text encoder to enable a more comprehensive, bidirectional co-optimization framework.
>
> [T1] MTEB: Massive Text Embedding Benchmark
>
> > **Q3** Role of IDF Weighting
>
> **A:** Thank you for the suggestion. We have conducted an ablation comparing IDF-weighted labels with unweighted K-hot labels. To speed up experiments without impacting conclusions, we used a ViT-B/16 model trained on 256M samples (vs. 512M in the paper), keeping all other settings unchanged.
>
> | Design   | Imagenet val | COCO T2I | Flicker T2I | COCO I2T | Flicker I2T |
> |----------|--------------|----------|-------------|----------|-------------|
> | IDF      | **47.1**     | **33.2** | **54.7**    | **48.9** | **73.1**    |
> | w/o IDF  | 44.8         | 31.6     | 51.7        | 48.0     | 71.1        |
>
> Above results show that IDF weighting improves performance across tasks, supporting its effectiveness. We will add these results in the revised version to highlight the benefit of IDF weighting.

---

> > ### Comment · Reviewer_Uu27 · 2025-08-05
> >
> > Thank you for the detailed response and the additional ablation experiments.
> >
> > In my opinion, a larger weight on L_{Class} ​corresponds to a relatively smaller weight on L_{CLIP} ​(the text-to-image and image-to-text retrieval loss). Intuitively, one might expect that reducing the influence of the CLIP-based retrieval loss could degrade retrieval performance. However, the ablation results show the opposite trend, which is somewhat counterintuitive. Could you please provide some insight into why a higher weight on L_{Class}​ does not harm, and may even improve retrieval performance? Thank you!

---

> ### Author Response · Authors · 2025-08-05
>
> Dear Reviewer,
>
> Thank you for your valuable comments and questions. We have done our best to address them in our responses. Please kindly let us know if there are any remaining concerns or points that need further clarification.
>
> Sincerely,
>
> SuperCLIP Authors

---

> ### Author Response · Authors · 2025-08-05
> **A larger weight on L_{Class} resulting in better retrieval performance**
>
> We appreciate your thoughtful observation. This seemingly counterintuitive result can be explained as follows:
>
> **1. Synergistic Effects of Alignment Objectives**
>
> Both L_{CLIP} and L_{Class} serve to improve image-text alignment, albeit through different mechanisms. While L_{CLIP} enforces global image-text correspondence, L_{Class} provides a complementary fine-grained supervisory signal by aligning image representations with caption-derived N-hot vectors. Increasing the weight of L_{Class} does not simply reduce the effect of L_{CLIP}; rather, it introduces additional discriminative information that can further refine the joint embedding space. As a result, retrieval performance may improve due to richer and more robust visual-semantic alignments.
>
> **2. Enhanced Dual-Encoder Representations**
>
> Retrieval tasks depend on the quality of both visual and textual encoders. While L_{Class} has a clear benefit for visual representations, our additional experiments (see our response to Q2: secondary effects on the text encoder's representations) show that it also improves text encoder performance. Specifically, the text encoder trained with SuperCLIP consistently outperforms the vanilla CLIP model across various text benchmarks, including semantic similarity, text classification, clustering, and retrieval. These improvements in the text encoder directly contribute to better image-text retrieval performance.
>
> In summary, increasing the weight of L_{Class} not only avoids harming retrieval performance but can enhance it, thanks to the complementary and mutually reinforcing effects on representation learning for both modalities. We hope this response addresses your question. Thank you again for your insightful feedback.

---

> > ### Comment · Reviewer_Uu27 · 2025-08-05
> >
> > Thank you for your response, I have no question now.

---

> > > ### Author Response · Authors · 2025-08-07
> > >
> > > We are grateful for your thorough review and helpful comments. Thank you.

---

### Official Review · Reviewer_6sBX · 2025-07-19

**Clarity:** 4
**Significance:** 2
**Originality:** 2
**Rating:** 4
**Confidence:** 4

**Summary:**

This paper addresses the CLIP model’s limited ability to capture fine-grained information by proposing a simple yet effective training strategy. Extensive experiments across diverse evaluation settings and backbone models demonstrate that the proposed method significantly enhances the model’s fine-grained image-text alignment capabilities.

**Questions:**

1. SuperCLIP is trained from scratch. Although the additional computational overhead is relatively small compared to CLIP's full pre-training, it’s worth considering whether the computational burden could be further alleviated by applying post-training strategies on top of a pretrained CLIP model. Especially since the introduction of a linear head does not alter CLIP’s original architecture.
2. How is inference performed in SuperCLIP? Are the predictions obtained by combining (e.g., weighting) the outputs from the linear head and CLIP’s original predictions?
3. As shown in Table 2, the pretrained version of B-512M appears to have different zero-shot performance compared to CLIP ViT-B/16. Are these two versions equivalent? If not, what are the key differences between them?
4. The authors use token frequency as a proxy for estimating the amount of lexical information, but this approach is somewhat coarse and imprecise.

**Ethical Concerns:**

["NO or VERY MINOR ethics concerns only"]

**Final Justification:**

The work is relatively complete and offers some degree of inspiration. However, my concerns have not been fully addressed — in particular, the authors’ responses to Q1 and Q3 are not entirely satisfactory. Therefore, I will keep my score unchanged.

**Limitations:**

The paper lacks a discussion of its limitations.

**Paper Formatting Concerns:**

No.

**Quality:**

3

**Strengths And Weaknesses:**

# Strength
1. The paper is well-structured, with a clear motivation that makes it easy to follow.
2. It introduces SuperCLIP, which leverages raw text tokens to enhance CLIP’s image-text alignment capabilities.
3. The evaluation is conducted across a wide range of scenarios and is not limited to the CLIP model alone.

# Weakness
See Questions.

---

> ### Author Rebuttal · Authors · 2025-07-31
>
> > **Q1** Post-training strategies to reduce computational cost on pretrained CLIP.
>
> **A:** Thank you for the insightful comment. Due to time constraints, we have not yet fully explored post-training strategies on a pretrained CLIP model.
>
> However, based on our previous experiments, we observed that adding the classification loss not only enhances visual representation but also accelerates convergence. This suggests that our method could potentially adapt a pretrained CLIP model to new tasks or data domains more quickly during a post-training phase. We plan to investigate this further in our future work.
>
> > **Q2**  Inference procedure and the role of the linear head in SuperCLIP.
>
> **A:**  Thank you for the question. Inference in SuperCLIP follows the standard CLIP procedure: predictions are made via cosine similarity between image and text embeddings. The added linear classification head is only used during training to enhance supervision for the vision encoder. It is not used at inference time, and its outputs are not combined with CLIP predictions. We will clarify this in the revised paper.
>
> > **Q3**  Equivalence between pretrained B-512M and CLIP ViT-B/16.
>
> **A:**  Thank you for your question. We are not entirely sure which version of CLIP ViT-B/16 you are referring to, so we will address both possibilities.
>
>   1. Compared to the open-source CLIP ViT-B/16 from OpenCLIP: We use the same model architecture and training dataset. The primary difference in accuracy stems from the number of seen samples. OpenCLIP achieves 73.5% on ImageNet-1K with 13B seen samples, whereas our model achieves 60.5% with 512M seen samples.
>
>   2. Compared to the original OpenAI CLIP ViT-B/16: While we use the same architecture, the performance difference is mainly due to different training datasets and the number of seen samples. The original OpenAI model achieves 68.6% on ImageNet-1K using the WIT-400M dataset with an undisclosed number of seen samples. Our model achieves 60.5% using the DataComp-1B dataset with 512M seen samples. Additionally, minor differences in training details, such as learning rate and batch size, may also contribute to the performance gap.
>
> In our experiments, we ensured fair comparisons by using the same training settings, including dataset and data volume, for both our proposed SuperCLIP and the baseline CLIP model.
>
> > **Q4**  Limitations of using token frequency to estimate lexical information.
>
> **A:**  Thank you for your question.
>
> In SuperCLIP, We adopt IDF-based token frequency as a proxy to estimate lexical information primarily to downweight common but less informative tokens. While simple, this method effectively highlights semantically meaningful words and helps guide visual attention. Our additional ablation study confirms that this strategy improves performance.  To speed up experiments without impacting conclusions, we used a ViT-B/16 model trained on 256M samples (vs. 512M in the paper), keeping all other settings unchanged.
>
> | Design   | Imagenet val | COCO T2I | Flicker T2I | COCO I2T | Flicker I2T |
> |----------|--------------|----------|-------------|----------|-------------|
> | **IDF**      | **47.1**     | **33.2** | **54.7**    | **48.9** | **73.1**    |
> | w/o IDF  | 44.8         | 31.6     | 51.7        | 48.0     | 71.1        |
>
> While we currently adopt token frequency as a proxy for lexical information, we plan to explore additional dimensions such as lexical diversity and semantic complexity in future work.

---

### Note · Authors · 2025-08-13

Dear PC, SAC, AC, and Reviewers,

Thank you for your valuable efforts and constructive feedback. We sincerely appreciate the recognition of our work’s contributions, including:
- **Simplicity, Elegance, and Practicality**: A simple, elegant method introducing a powerful supervisory signal with negligible computational overhead, making it highly practical and scalable (Reviewer Uu27).
- **Superior Performance and Rigorous Evaluation**: Consistent outperformance of CLIP across multiple benchmarks, validated by a comprehensive and rigorous evaluation (Reviewers 6sBX, Uu27, jiRy, 2yeS).
- **Addressing Key CLIP Limitations**: Successfully mitigating CLIP's well-known sensitivity to batch size, a significant practical contribution (Reviewer Uu27).

During the rebuttal and discussion phases, we provided point-by-point responses to address the concerns. We are pleased that our clarifications successfully resolved all concerns for Reviewers jiRy, Uu27, and 2yeS. We also addressed the majority of concerns from Reviewer 6sBX and appreciate the maintained positive evaluation for the overall completeness and quality.

**Key resolutions and additions include**:
- Comprehensive ablation studies: Following reviewers' suggestions (Uu27, 2yeS, 6sBX), we performed extensive ablations on loss weighting and IDF weighting, confirming the robustness and effectiveness of our design choices.
- Strengthened comparisons with related works: We added new baselines like DetailCLIP (as suggested by jiRy) and expanded evaluations on vision-language compositionality benchmarks (per 2yeS), demonstrating SuperCLIP's superior performance and simplicity.
- Broadened evaluation scope and applicability: To address questions on broader impact (jiRy, Uu27), we evaluated the text encoder on MTEB and successfully integrated SuperCLIP into LLaVA-1.5, showing consistent improvements in diverse multimodal tasks.
- Improved clarity and deepened analysis: We clarified key aspects of our methodology, such as the inference process (per 6sBX), and provided deeper analysis on the model's robustness to rich textual signals (as requested by jiRy).

All feedback from the rebuttal phase will be incorporated into the revised manuscript. We will also open-source our code and models to ensure reproducibility and facilitate future research.

We sincerely thank all reviewers, the AC, SAC, and PC for their time, effort, and constructive input, which have been invaluable in helping us improve the work.

Best,

Authors

---

### Decision · Program_Chairs · 2025-09-17

**Decision:**

Accept (poster)

**Comment:**

Overall, the paper proposes SuperCLIP, a simple yet effective modification to the CLIP model that enhances fine-grained image-text alignment by adding a lightweight linear layer and a multi-label classification loss based on raw text tokens. Reviewers appreciated the paper’s clarity, simplicity, and thorough evaluation across diverse tasks, including zero-shot classification, image-text retrieval, and visual benchmarks. While some concerns were raised regarding computational efficiency, sensitivity to loss weighting, the one-sided supervision flow, and the limited comparison to other baselines, the paper was recognized for its practical impact, rigorous experimental validation, and potential to inspire future research in improving vision-language models.

During the rebuttal, the authors adequately addressed reviewers’ concerns by clarifying implementation details, experimental settings, and the design choices behind SuperCLIP. Remaining minor questions did not significantly detract from the technical contribution or the empirical results. Given the paper’s solid methodology, consistent performance gains across benchmarks, and practical significance, the final recommendation is to accept the paper.